# Male Involvement in Family Planning Decisions in Malawi and Tanzania: What Are the Determinants?

**DOI:** 10.3390/ijerph20065053

**Published:** 2023-03-13

**Authors:** Godswill Nwabuisi Osuafor, Monica Ewomazino Akokuwebe, Erhabor Sunday Idemudia

**Affiliations:** 1Department of Population Studies and Demography, North-West University, Mafikeng 2735, South Africa; 23376430@g.nwu.ac.za; 2Faculty of Humanities, North-West University, Mafikeng 2735, South Africa; erhabor.idemudia@nwu.ac.za

**Keywords:** determinants, family planning decisions, Malawi, male involvement, prevalence, Tanzania

## Abstract

The participation of males in joint spousal decisions is urgently needed in achieving the fundamental indicators of reproductive health. The low involvement of males in family planning (FP) decision-making is a major determining factor in low FP usage in Malawi and Tanzania. Despite this, there are inconsistent findings regarding the extent of male involvement and the determinants that aid male participation in FP decisions in these two countries. The objective of this study was to assess the prevalence of male involvement in FP decisions and its associated determinants within the household context in Malawi and Tanzania. We used data from the 2015–2016 Malawi and Tanzania Demographic and Health Surveys (DHSs) to examine the prevalence and the determinants inhibiting male involvement in FP decisions. The total sample size of 7478 from Malawi and 3514 males from Tanzania aged 15–54 years was employed in the analysis by STATA version 17. Descriptive (graphs, tables and means), bi-variate (chi-square) and logistic regression analyses (unadjusted (U) and adjusted odds ratio (AOR)) were performed to identify the determinants associated with male involvement in FP decisions. The mean age of respondents in Malawi was 32 years (±8 SD) and in Tanzania, 36 years (±6 SD), with the prevalence of male involvement in FP decisions being 53.0% in Malawi and 26.6% in Tanzania. Being aged 35–44 years [AOR = 1.81; 95% CI: 1.59–2.05] and 45–54 years [AOR = 1.43; 95% CI: 1.22–1.67], educated (secondary/higher) [AOR = 1.62; 95% CI: 1.31–1.99], having access to media information [AOR = 1.35; 95% CI: 1.21–1.51] and having a female head of household [AOR = 1.79; 95% CI: 1.70–1.90] were determinant factors of male involvement in FP decisions in Malawi. Primary education [AOR = 1.94; 95% CI: 1.39–2.72], having a middle wealth index ranking [AOR = 1.46; 95% CI: 1.17–1.81], being married [AOR = 1.62; 95% CI: 1.38–1.90] and working [AOR = 2.86; 95% CI: 2.10–3.88] were higher predictors of male involvement in FP decisions in Tanzania. Increasing the role of males in FP decisions and involvement in FP utilization may improve uptake and continuity of FP usage. Therefore, the findings from this cross-sectional study will support redesigning the ineffective strategic FP programs that accommodate socio-demographic determinants that may increase the likelihood of male involvement in FP decisions, especially in the grassroots settings in Malawi and Tanzania.

## 1. Introduction

Family planning (FP) service use has been found to be particularly effective in preventing unintended pregnancies and drastically led to the reduction of child and maternal mortality burden in low- and middle-income countries. In order to improve maternal health, a significant public health initiative is to increase male participation in FP decision-making [1,2,3,4]. Therefore, FP is a conscious effort made by couples to decide how many children they want to have and when they want to have them [3,5]. It aims at improving family lives and contributing to sustainability efforts at both the micro and macro levels. FP is a method of population control and enhances the reduction of unplanned pregnancies by providing direct or indirect benefits, such as lowered maternal morbidity and mortality [5,6], increased educational and employment opportunities for women who are able to delay childbirth [7,8], decreased use of unsafe abortion methods [9], reduced risk of HIV transmission to unborn children [10,11], and lowered neonatal, infant, and child mortality [12]. According to the World Health Organization (WHO) [13], after years of concerted work, a significant proportion of women worldwide wish to avoid getting pregnant, but neither they nor their partners are using contraceptives. This unmet need is due to a number of factors, including the availability of FP services, the dearth of a choice of FP procedures, the fear of partner opposition, and concerns about side effects and health issues, among many others [13].

One of the major factors that deserves attention is the involvement of males in decisions about the uptake of family planning methods. Males are typically not the primary decision-makers in African families, despite the fact that they are crucial to the acceptability of FP approaches, according to previous studies [8,14]. Conversely, male acceptance of the use of FP methods still remains low in sub-Saharan African countries [4,13]. In order to achieve the core indicators of reproductive health, it is essential to encourage males to require joint spousal decisions, as this approach will allow couples to plan their families in accordance with their resources and needs. On the other hand, religious barriers, lack of male involvement, and traditional beliefs have led to weakened FP interventions put in place to curb the burdens of pregnancy [15,16]. Several studies have focused on the importance of FP usage; however, most of these FP studies carried out in developing countries are primarily centered on women and their use of FP methods [17,18]. Therefore, the roles and responsibilities of men participating in FP and fertility regulation have been ignored, understudied, and underutilized. For instance, it was discovered that Kenya’s program attempts to encourage male involvement in reproductive health and FP services were minimal [19].

The Family Association of Kenya (FPAK) implemented several projects, including the Males as Partners project and Young Men as Equal Partners project [20,21,22,23], to increase male participation in FP in response to the International Conference on Population and Development (ICPD) held in Cairo in 1994 [20,21]. The ICPD’s program of action highlighted that further efforts should be made to incorporate men’s shared responsibilities and to foster their active involvement in responsible parenting, sexual, and reproductive issues [21]. At the 1995 Beijing World Women’s Conference, messages stressing the “shared responsibility of men and women in all problems relating to reproductive health” were once again emphasized. The advancement of male participation in FP and reproductive health matters will foster the promotion of the health of their families and assist in helping to achieve equity in sexual behaviours. The low prevalence of contraceptive use in sub-Saharan Africa and other low-income countries [24,25] is a consequence of gender dominance, particularly men’s negative evaluations of FP. With this said, the advantages of male participation in FP services are increasingly being acknowledged. For instance, a study in Bangladesh found that 40% of men were involved, and another in western Nigeria found that 39.6% were [26,27]. Nonetheless, this demonstrated that despite ongoing efforts towards wider coverage of FP services, male involvement in FP decision-making with their partners is still low. According to a study in Nigeria, where 62% of women cited their husbands as their decision-makers and only 6% of currently married women at the time of the survey made decisions for themselves [28], the effect of male dominance on the decision-making process exacerbates indicators of poor reproductive health.

Male participation in this context should be interpreted much more broadly than male contraception, which has the impact of raising the acceptance and prevalence of FP practices of either sex [29,30]. This is a fact for families in Malawi and Tanzania, where patrilineal traditional norms that are typically dominated by men determine how society functions. An unmet need for FP is one of the many reasons Malawi is among the nations in Africa with the highest maternal mortality ratio (MMR) and fertility rate (FR) [30,31]. Due to Malawi’s perception of FP as a women’s domain, male participation in FP is still less than desirable. With programs to increase the use of contraceptives, known as the Malawi Male Motivator Project [31], men were encouraged to use FP techniques. In comparison to other countries, Tanzania has one of the highest total fertility rates (TFR), at 5.2, and a level of FP usage of 32% [32]. Low involvement of men in FP is one of the factors influencing its utilization in Tanzania, owing to male dominance in FP decision-making. Even though FP is known to prevent maternal deaths, cultural norms and service delivery factors have been cited in studies that create family programs which are difficult to achieve in Tanzania [32,33,34,35]. Despite the fact that awareness of its use is now largely universal, about 34% of married women in Tanzania utilized any kind of contraception. Similarly, it was discovered that only 33% of people in the nation understood periodic abstinence and that only 20% of women and 14% of males knew the right timing of a fertile period [32,36]. Furthermore, Tanzania has some of the highest maternal death rates in sub-Saharan Africa. According to the 2015–2016 Tanzania Demographic and Health Survey, there are 556 maternal deaths for every 100,000 live births, with rural areas having a higher rate than urban ones [32,37].

Men are frequently regarded as the primary decision-makers regarding family size and the use of FP, and therefore, there is an urgent need to increase FP uptake in Malawi and Tanzania. Several studies in high- and middle-income countries have identified a strong association between male involvement and an upsurge in contraceptive uptake and use [19,38]. Men who permit shared spousal decisions must be included if the reproductive health indicators are to be met. One of the factors contributing to the high rates of maternal illness and mortality is the low degree of male participation in reproductive health practices. This has decreased the effects of FP interventions and coupled with uncontrolled fertility, which impedes economic growth and throws a nation’s politics into instability [39,40]. Studies on FP have found that male involvement is positively correlated with access to the media, including television and radio, spouse employment status, and average monthly income [41,42]. Despite this, there are conflicting results regarding the level of male involvement in FP decisions and the factors that influence this involvement in Malawi and Tanzania. As a result, there is a dearth of information regarding the factors that influence male involvement in FP services in these countries. In order to help policymakers build FP programs that will promote male involvement in FP choices, data must be generated. Therefore, the purpose of this study was to determine the point prevalence of male involvement in FP decisions in Malawi and Tanzania, as well as the factors that influence this involvement. Additionally, this study seeks to contribute to the body of knowledge by exploring male involvement in FP decisions within the household context, using the demographic health surveys.

## 2. Methods

### 2.1. Data Source and Sampling Procedures

The demographic health survey (DHS) data of Malawi and Tanzania were used for this study (the 2015–2016 Malawi Demographic and Health Survey (2015–2016 MDHS); Tanzania Demographic and Health Survey (2015–2016 TDHS)) [43,44]. We used the most recent DHSs from each country as secondary data sources. These surveys are available through the DHS Program website. The DHS Program provides on-request public access to their data via an application programming interface (API), from which microdata for each country could systematically be downloaded. Further details regarding the DHS survey methodology and complex sampling can be reviewed on the DHS Program website (https://dhsprogram.com/methodology/ (accessed on 14 June 2022)). The DHS, a nationally representative survey, collects information on health and factors related to it, such as mortality, morbidity, use of family planning services, fertility, and maternal and child health. In short, DHSs follow standardized data collection procedures by employing similar questionnaires across different countries, allowing comparability between countries regarding the variables specifically studied (The DHS Program, 2022) [45]. In order to ascertain the point prevalence and contributing factors of male involvement in FP decisions in Malawi and Tanzania, the variables were taken from the literature and added together. The DHS employed a two-stage stratified sampling technique to select the study respondents. In the first stage, enumeration areas (EAs) were randomly selected, while in the second stage, households were selected. Each country’s survey consists of different datasets including men, women, children, birth, and household datasets, and for this study, we used the men’s datasets (MR file). This study included a weighted sample of 10,992 men aged 15–54 who were sexually active, knowledgeable about FP methods, and more likely to be involved in FP decisions or to have had prior experience with being involved in FP decisions in the five years prior to the survey. Regarding the limitations of the DHS datasets, these include reporting and recall bias, particularly for retrospective data relying on memory of a past event.

### 2.2. Study Variables and Measurements

#### 2.2.1. Outcome Variable

The outcome variable for this study was male involvement in FP decisions. Men who in the five years preceding the survey had used any contraceptive methods (traditional and modern methods), or had knowledge that using condoms does not decrease men’s sexual desire, or believed that men should not care about contraception as it is a woman’s responsibility, or thought that having too many children was often detrimental to the mother’s health, or thought that men should share FP practices in the family, were selected. Male involvement in FP decisions was categorized into ‘Yes’ (involvement in FP decisions) or ‘No’ (non-involvement in FP decisions). The response variable for the ith male was represented by a random variable Yi, with two possible values coded as 1 and 0. Thus, the response variable of the ith male Yi was measured as a dichotomous variable with possible values Yi = 1, if ith man discussed FP with health workers or health professionals, and Yi = 0 if the male never discussed FP with health workers or health professionals in the last few months preceding the survey.

#### 2.2.2. Independent Variables

The independent variables retrieved from the DHS were age, place of residence, education, wealth index, marital status, occupation, exposure to media, contraceptive knowledge, and the sex of the household head (Table 1). The years of the surveys were decided upon as an independent variable by using 2015–2016 as a reference because the DHSs of the countries of Tanzania and Malawi were taken into consideration at the same time. The years of the surveys were classified as 2015–2016 (Malawi) [43] and 2015–2016 (Tanzania) [44]. However, the bi-variable analysis with a *p*-value of > 0.2 were not eligible to be included in the multivariable analysis.

### 2.3. Data Analysis

STATA version 14 statistical software was used for data management and analysis. First, descriptive statistics were used to provide sample characteristics of the respondents. Bar graphs were used for the illustration of the point prevalence of male involvement in FP decisions in Malawi and Tanzania. Second, bi-variate analysis was performed using the Chi-Square (χ^2^) test statistic to define the statistical relationship of the outcome and the explanatory factors. Third, multivariate analysis (binary logistic regression) was performed to test the determinants associated with male involvement in FP decisions, which significantly predicts the outcome variable. The binary logistics regression model assesses the effect of socio-demographic factors on male involvement in FP decisions in a multiple regression framework. Following Tolles et al. (2016), the binary logistic regression model is defined as:$$\frac{\pi }{1-\pi }=exp(\beta ο+\beta^{1}X^{1}+\cdots +\beta \kappa X\kappa )$$
which is an equation that describes the odds of being in the current category of interest and by definition, the odds for an event are *π*/(1 − *π*) such that *p* is the probability of the event. Thus, the multivariable logistic regression analysis took into account variables that had a *p*-value of less than 0.2 in the bivariate analysis. The unadjusted odds ratio (UOR) and the adjusted odds ratio (AOR) with 95% confidence interval (CI) were reported to declare the statistical significance and strength of association between the predicting determinants and the outcome variable (*p* < 0.05) in the multivariable logistic regression model. All analyses were weighted to account for differences in sampling probabilities.

### 2.4. Ethical Considerations

This study employed freely-accessible unidentified datasets, which suggests that the datasets themselves were not identified, rather than the respondents. One of the authors (corresponding author-Monica Ewomazino Akokuwebe) requested approval from the DHS Program/ICF International to download and use the dataset for all the countries analyzed in the study.

## 3. Results

### 3.1. Socio-Demographic Characteristics of the Respondents in Malawi and Tanzania

Table 2 presents the socio-demographic factors of the respondents in Malawi and Tanzania. A total of 7478 males from Malawi and 3514 male respondents in Tanzania, totaling 10,992 male respondents altogether, were included in the study. The mean age of respondents in Malawi was 32 years (±8 SD) and in Tanzania, 36 years (±6 SD). In both countries, a majority of respondents were found to be in the 15−24-year-old age group and are living in rural places of residence. About 56.9% and 61.9% respectively reported having primary education, and the majority of them were found to have rich wealth status (Malawi 49.1%; Tanzania 47.3%) (Table 2).

### 3.2. Prevalence of Involvement in FP Decisions by Countries

The point prevalence of male involvement in FP decisions in Malawi and Tanzania is shown in Figure 1. The bar graph demonstrated that male involvement in FP decisions was at a point prevalence of 53.0% in Malawi and 26.6% in Tanzania (Figure 1).

### 3.3. Male Involvement in FP Decisions Associated with Characteristics of the Respondents

Table 3 presents the bi-variate analysis of male involvement in FP decisions in Malawi and Tanzania. In Malawi, 60.3% of the males aged 25–34 years reported being involved in FP decisions (*p* < 0.05), while 58.8% of the males with secondary/higher education reported being involved in FP decisions (*p* < 0.05). Also, 53.7% of the males who were Christians indicated their involvement in FP decisions (*p* < 0.05), while 66.4% of those divorced/separated, as well as 54.7% of those who had access to media information reported their involvement in FP decisions (*p* < 0.05). In Tanzania, 34.5% of male respondents who were aged 35–44 years reported being involved in FP decisions (*p* < 0.05), while 32.5% of them living in urban places of residence reported being involved in FP decisions (*p* < 0.05), and 30.5% of them indicated rich wealth status (*p*<0.05). Also, about 47.1% them who are widowed reported their involvement in FP decisions (*p* < 0.05), and 28.5% of them working reported being involved in FP decisions (*p* < 0.05) (Table 3).

### 3.4. Determinants Associated with Male Involvement in FP Decisions

Table 4 presents the findings of the multivariate logistic regression analysis, estimating the adjusted odds ratio (AOR) in relation to the factors predicting male involvement in FP decisions in Malawi and Tanzania. In Malawi, having access to media information, being between the ages of 35–44 and 45–54, having secondary or higher education, having a rich wealth status, practicing either Islam or Christianity, and having a female head of household were all significantly associated with male involvement in FP decisions. In Tanzania, determinants such as residing in a rural residence, having primary education, being of middle wealth status, working, and having a female head of household were found to be significantly associated with male involvement in FP decisions in the multivariate logistic regression analyses. Males in Malawi who were aged 35–44 years [AOR = 1.81, 95% CI: 1.59, 2.05] and 45–54 years [AOR = 1.43, 95% CI: 1.22, 1.67] had higher odds of being involved in FP decisions compared to males aged 15–24 years. Males who attained secondary/higher education had 1.62 times [AOR = 1.62, 95% CI: 1.31, 1.99] higher odds of being involved in FP decisions compared to males who had no education (Table 4). Male respondents from a household with rich wealth status had 1.18 [AOR = 1.18, 95% CI: 1.07, 1.31] times higher odds of being involved in FP decisions than males from a poor household. The odds of involvement in FP decisions by males who were Muslim were lowered by 82% [AOR = 0.82, 95% CI: 0.70, 0.95], and for those of traditional religion, their odds were lowered by 70% [AOR = 0.70, 95% CI: 0.52, 0.93] compared to Christian males in Malawi (Table 4). Male respondents with access to media information had 1.35 [AOR = 1.35, 95% CI: 1.21, 1.51] times higher odds of being involved in FP decisions compared to males who did not have access to media information. Male respondents in Malawi with a female head of household had 1.79 [AOR = 1.79, 95% CI: 1.70, 1.90] times higher odds of being involved in FP decisions compared to those with a male head of household (Table 4).

In Tanzania, the multivariate logistic regression model showed that rural residence, primary education, middle wealth status, being in work, and the head of household being female were significantly associated with male involvement in FP decisions. Male respondents had 0.66 [AOR = 0.66, 95% CI: 0.57, 0.78] times lower odds of being involved in FP decisions compared to their urban counterparts (Table 4). Those with primary education had 1.94 times [AOR = 1.94, 95% CI: 1.39, 2.72] higher odds of being involved in FP decisions than those with no education. Respondents with middle wealth status had 1.46 times [AOR = 1.46, 95% CI: 1.17, 1.81] higher odds of being involved in FP decisions compared to those with poor wealth status (Table 4). The odds of being involved in FP decisions by respondents who were working were 2.86 times [AOR = 2.86, 95% CI: 2.10, 3.88] higher than males who were not working, and males who had a female head of household had 0.80 times [AOR = 0.80, 95% CI: 0.64, 1.01] lower odds of being involved in FP decisions than males who had a male head of household in Tanzania (Table 4).

## 4. Discussion

This study was performed to ascertain the extent of male involvement and its contribution to determining FP decisions among men in Malawi and Tanzania, in southern and east Africa. We found out that the prevalence of male involvement in FP decisions in Malawi and in Tanzania is 53.0% and 26.6%, respectively. The study’s findings showed that slightly more than half of respondents participated actively in FP decision-making in Malawi compared to Tanzania (26.6%). In contrast, a study from Ogun State, Nigeria, found that the prevalence of male involvement was 30.9% [27]. Additionally, other studies from Ghana (34.5%) [46], Ethiopia (44%) [47], and Bangladesh (40%) [48] found higher rates of male involvement in FP decisions than in Tanzania. Notwithstanding the fact that a prior study in an urban municipality in Bangladesh found a male involvement rate of 63.2% in FP decisions [49], a recent study in Ethiopia indicated a greater prevalence of male involvement (68%) in FP decisions than by males in Malawi and Tanzania [50]. The dearth of male FP methods [48], the patriarchal societies that exist in the African environment [40], and the pervasive myths [51] and misconceptions [52] connected with FP use could all be seen as reasons for the low participation of men in FP decisions. Enhancing men’s FP services and working to dispel widespread misconceptions regarding FP facilities are also extremely important.

The findings of this research found that men who are aged 25–34 (60.3%), with secondary/higher education (58.8%), who are rich (54.9%), are Christians by religion (53.7%), divorced/separated (66.4%), working (56.2%), with access to media information (54.7%), with a good knowledge of contraceptives (53.3%), and in households headed by males (53.9%), were most involved in FP decisions in Malawi. In Tanzania, men aged 35–44, who reside in urban areas (32.5%), have secondary/higher education (29.4%), are rich (30.5%), widowed (28.2%), working (28.2%), have access to media information (28.8%), have a good knowledge of contraceptives (27.3%), and reside in households headed by males (27.2%), reported greater male involvement in FP decisions. This is consistent with prior studies on male involvement in FP decisions conducted in India [53], Uganda [54], Kenya [55], and Ethiopia [48]. Importantly, studies have shown that condom usage was low among men who used contraception, indicating that both the use of other male contraceptive methods and men’s participation in FP decisions were extremely low. This could be due to the fact that men only have a handful of FP services to choose from. Hence, efforts should be reinforced to educate males with options for permanent FP for those who are definitely certain that they do not want children in the future. Male interest in FP should, however, go beyond the use of contraceptive methods, and rather emphasize male participation, which can be assessed in the present study by spousal communication and approval [3,56]. Studies have revealed that male involvement in FP decisions can be made possible if spousal communication and approval is encouraged in FP discussions among couples. Other studies conducted in Kenya [57], Cameroon [58] and Ethiopia [59] confirmed the aforementioned assertion of spousal communication and approval of couples jointly engaged in FP decisions. The conceptualization of the idea of male involvement in FP decisions, which changes greatly among study and research countries, is most likely the basis of the effect’s inconsistency; hence, interpretation of the study’s findings ought to be carried out with consideration.

Our study identified several determinants associated with male involvement in FP decisions in Malawi and in Tanzania. Determinants of male involvement in FP decisions in this study were attributed, in Malawi, to age (35–44 years and 45–54 years), primary and secondary/higher education, middle and rich wealth status, being divorced/separated, working, access to media information, and being in a male-headed household. In Tanzania, determinants including age (35–54 years), secondary/higher education, middle and rich wealth status, being widowed, working, and having access to media information were predictors of male involvement in FP decisions. Other studies have come out with similar findings [25,41,42]. Male decision-making regarding FP is influenced by the socio-cultural context, including individual views and feelings towards male involvement in FP use and decision-making. Most men in rural areas of underdeveloped nations can be reluctant to insist that their wives follow FP practices they are unfamiliar with. As shown by studies, some men stand in opposition to the use of FP methods owing to tradition and religion beliefs, which allows males to maintain their power over women within their extended families, villages, social groups, and religious organizations [30,60]. Similar results have been found in studies conducted in rural northern Ghana [60,61]. Men are deterred from taking their wives to health clinics to discuss FP issues because of the intricate web of social and cultural barriers that obstruct spousal communication about reproductive health issues [40,62,63].

Males’ participation in FP decisions with their spouses may vary depending on the level of education they have attained. Educated men would know the benefit of being involved in FP decisions, which creates an avenue for spousal communication and gathering information on FP services through reading newspapers, mass media, and from different social media [64,65]. Generally, when educated males have access to media information, their odds of participating in FP and reproductive health services are increased [66,67]. Therefore, government and civil society organizations, FP program planners, and various stakeholders should ensure availability, accessibility, and sustained advocacy for male participation in FP decision-making and the effective utilization of FP services. This should be in addition to creating a conducive environment for the dissemination of knowledge on the tremendous advantages of male participation in FP decisions among couples, behavioral change, and open discussion on matters relating to females’ reproductive health, including FP methods and services [66,68]. This study finding confirms the outcomes from earlier research that suggested communication might help males participate in FP choices.

Since men have historically been seen as being uninvolved in fertility outside of their ability to supply sperm to women for reproduction, the primary focus of FP programs has generally been on women, frequently to the exclusion of men [68]. Men’s involvement in FP programs should be properly assessed not only in terms of FP outcomes, but also in terms of how it improves social norms and appropriate relationships between males and females in FP decisions. In order to encourage male participation in FP decisions alongside their partners, there are a number of national FP programs that are funded by governments and key stakeholders [68,69]. For instance, the Johns Hopkins Bloomberg School of Public Health’s Center for Communications Programs’ Male Motivation Campaign was a Behaviour Change Communication (BBC) intervention of the PRISM Project in Guinea. The immediate objectives of this campaign were to promote access to FP and other related healthcare services, raise demand for them, improve the quality of care, and enhance coordination and links between healthcare professionals and services [68,70]. This BCC campaign was carried out in two stages: the first stage employed advocacy interventions to increase support for FP among religious leaders, while the second stage concentrated on married men.

The usage of services was increased by the use of the multimedia interventions that encouraged discussion between spouses regarding FP. The Male Motivation Campaign, on the other hand, was introduced to secondary audiences, such as women of childbearing age and service providers, and it was made available nationally as well as in three project regions [64,71]. The campaign included a radio soap opera called “*La Vie Nést Pas Compliquee*,” radio programs in the local tongue, radio spots, and entertainment education cassettes using humor in their messages, traditional music contests, community mobilization events, and billboards, posters, brochures, and promotional materials for men [65,68]. Additionally, activities and materials varied for men, women, and religious leaders. Thus, women were reached out to through radio programs, radio spots, and community mobilization activities, while religious leaders saw a movie in their own tongue of religious leaders debating FP and other reproductive health (RH) concerns against the backdrop of Islamic texts.

Moreover, seminars for Muslim and Christian religious leaders were organized, along with the compilation of an FP booklet and a poster [64,71]. Moreover, Kenya’s national FP campaign was created to reach both men and women, and it tends to center on the obvious facts of population growth in order to justify the necessity for FP. The initiatives also seek to demystify contraceptive techniques and assure the public of their efficacy and safety, particularly in rural regions where mistrust and misunderstanding are prevalent [68]. Similarly, FP is promoted in Senegal under the name “*Moytu Nef*”, which means to avoid having too many children at once. The campaign aims to empower women to make decisions about their number of children, and their timing, by educating them on the research on women’s health and birth spacing [67,68]. To support FP messages and assist them in reaching more target male audiences and addressing more unmet needs within families, the FP initiative even included male spouses and religious leaders.

Furthermore, in the southern Africa countries, there was a significant association between the wealth index and the likelihood that men would participate in FP decision-making. This study provides evidence that men with middle and high wealth status were more likely than poor men to participate in FP decisions. This finding was supported by studies conducted in Ethiopia [66] and Togo [40]. The explanation could be that men with medium and high household wealth indexes are more likely to be able to cover costs like transportation and have easy access to information about the advantages of participating in FP decisions. Yet, most research has revealed a high correlation between women’s socioeconomic position and their use of FP, and it has been proven that directly promoting and raising women’s economic status is an effective public health approach for enhancing maternal and child health outcomes [72,73,74]. Therefore, it is important to realize that the socioeconomic inequalities may exist in male involvement related to discussion and negotiation of FP issues between both partners, and hence, addressing socioeconomic empowerment of female spouses is equally important [75,76].

Additionally, this study found that the involvement of men in FP decisions is significantly predicted by men’s knowledge of contraceptives. Males who are knowledgeable about contraception are more likely than their counterparts who lack this knowledge to participate in FP decisions. This finding was supported by studies conducted in Tanzania [77] and Ethiopia [66]. This might be because knowledge of available contraceptive methods enables individuals to make informed decisions and use contraception to plan, delay, and space the pregnancies of their spouses [78], which is linked to improved birth outcomes for the mother and the newborns, either directly or indirectly. The sex of the household head is strongly correlated with the presence of men in FP decision-making. Households headed by males in Malawi have a higher likelihood of male involvement in FP decisions than their counterparts in Tanzania. This is in line with research conducted by Vouking et al. [3]. A possible justification for this might be because males are the major economic providers in the households, and are traditionally decision-makers. Marital status of respondents was positively associated with male involvement in FP decisions.

The current study showed that males who were divorced or separated had increased odds of involvement in FP decisions in Malawi, while males who were widowed were found to have increased odds of participation in FP decisions in Tanzania. A study carried out in Ethiopia [79] backed up this conclusion. This could be because joint decision-making between male and female partners to use contraception is a predisposing factor for male involvement in FP decisions among respondents who are married, cohabiting, widowed or divorced/separated. The joint participation in FP decision-making may exert a positive influence in meeting their reproductive health goals together as couples. The main strength of the present study was to present the findings from both countries using nationally representative data (most recent DHS), which improves the generalizability of the findings. Secondly, the response rate was greater than 90%. However, the study had limitations as well. First of all, because the study was cross-sectional, it cannot demonstrate a causal link between the predicted factors and the intended outcome. Secondly, the DHS’s use of an interviewer-administered form of data gathering instrument raises the possibility of bias.

## 5. Male Involvement in Family Planning and Its Implications for 2030 Agenda for SDG 3

Ensuring that everyone has access to family planning is one of the most effective and economical Sustainable Development Goals (SDG) targets, and a top priority for attaining the SDGs. Men are frequently the main decision-makers for family planning, despite the fact that contraceptive methods and services are primarily targeted at women. Male involvement in FP has become part of the major agenda in the Sustainable Development Goal (SDG) 3. It is now necessary to actively encourage fathers and male community members to help care for women and support their family in receiving adequate and improved healthcare. There is a positive correlation between male participation in family planning decisions and improved maternal and child health outcomes, according to a number of studies [80,81]. To achieve universal access to reproductive health care services, as part of the 2030 Agenda for Sustainable Development, it is crucial to increase access to contraception and ensure that the demand for FP is satisfied when utilizing effective contraceptive methods. Despite numerous coordinated efforts and success in increasing access to contraception, there are still significant obstacles to overcome [79,80]. Women’s empowerment, health, and gender equality are some of the related 17 SDGs goals/targets and the 2030 Agenda. Notwithstanding that family planning is crucial to achieving most of the objectives, it is specifically mentioned in SDG 3 on ensuring everyone’s health and well-being, and in SDG 5 on advancing women’s empowerment and gender equality for both men and women. Additionally, socio-cultural factors like stereotyping and the feminization of family planning have been mentioned in numerous studies as having the potential to prevent men from participating in FP discussions with their spouses [82,83].

In poor nations, the role of male involvement in family planning (FP) and their participation enhances women’s adoption of FP techniques, strengthens spousal coordination, helps the achievement of FP programs, and provides their partners with reproductive health rights [84]. Thus, the emphasis of men’s shared responsibility and the promotion of their active involvement in the prevention of sexually transmitted diseases, responsible parenthood, sexual and reproductive behaviour, including FP, and prevention of unwanted and high-risk pregnancies has become a major agenda in SDG 3 [79,85]. Involving men in FP services implies that there will be great improvement of health service utilization and this will ensure universal access to sexual and reproductive healthcare services, which include FP, education and information, and the integration of reproductive health into national strategies and programs. As a way to redefine the global strategy for achieving universal access to sexual and reproductive healthcare services and to identify potential avenues for both men and women’s involvement and their parity, countries should strengthen their national statistical capacities to compile national data for male involvement in FP. This will prepare regional and global aggregates, analyze implementation progress, and increase compliance with internationally agreed standards [40,84].

This will ensure advancement and better access to national data that will support increased investments in male involvement in FP decisions, particularly in sub-Saharan African nations where male involvement in FP decisions with their partners is still low. The socio-cultural barrier alone discourages men from participating in FP programs, so raising health literacy about FP among men should be included in the school curriculum to raise awareness of initiatives and counselling that will have a positive impact on men and encourage them to accept FP services and shared decision-making. Communication strategies on FP decisions should be encouraged among men and women to bring the benefits of FP and the rights of reproductive health for the realization of SDG 3.

## 6. Conclusions

In conclusion, the prevalence and total pooled prevalence of male involvement in FP decisions in Tanzania was significantly lower than in Malawi. Among the significant predictors of male involvement in FP decisions were older age, education, wealth status, occupational status, access to media information, and households having a male head of household, which are factors equally important to the involvement of males in FP decisions. Additionally, there may be other social factors that influence the variations in the prevalence and overall pooled prevalence of male involvement in FP decisions in Tanzania and Malawi. Thus, in Malawi, social problems, such as high HIV/AIDS infection rates, slow economic growth owing to COVID-19, and energy shortages, have over time posed additional challenges to the country. Similarly, Tanzania is faced with income poverty, and people suffer deprivation in several areas including health, nutrition, water, sanitation, housing, education, child protection, and access to information. Therefore, the use of FP services by men should be constantly taken into account in FP programs, because access to the media has a flow and dissemination power in sub-Saharan Africa. Policymakers and other stakeholders should take into account relevant laws and programs that assist spouses in making coordinated financial decisions. To reach more of the target male populations, it is advised to use FP campaign programs to spread awareness of sexually transmitted diseases (STDs). This study does not overstate the importance of FP as the best option for sustaining population growth, but it did take into account FP as a potential tool for improving mother and child health, as well as child survival rates.

## Figures and Tables

**Figure 1 ijerph-20-05053-f001:**
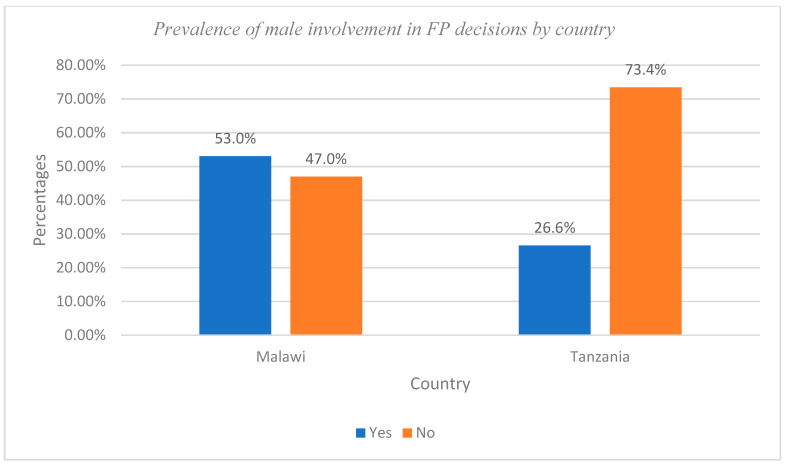
Prevalence of male involvement in FP decisions in Malawi and Tanzania.

**Table 1 ijerph-20-05053-t001:** The lists of independent variables and their definitions and measurements.

Variable Name	Definition (Measurement)
Age	Categorized as: 15–24, 25–34, 35–44, and 45–54 years
Place of residence	Categorized as: “rural” and “urban”
Education	Categorized as: “no education”, “primary education”, and “secondary/higher”
Wealth status	Categorized as: “poor”, “middle” and “rich”
Religion	Categorized as: “Christian”, “Muslim”, and “Traditional”
Marital status	Categorized as: “never married”, “married/cohabiting”, “widowed” and “divorced/separated”
Occupation	Categorized as: “not working” and “working”
Access to media information	Categorized as: “no” and “yes”
Contraceptive knowledge	Categorized as: “poor” and “good”
Sex of household head	Categorized as: “male-headed” and “female-headed”

**Table 2 ijerph-20-05053-t002:** Socio-demographic characteristics of the respondents.

Socio-Demographic Factors	Malawi, *n* = 7478	Tanzania = 3514
Age (in years)		
15–24	3226 (43.1%)	1556 (44.3%)
25–34	1975 (26.4%)	900 (25.6%)
35–44	1485 (19.9%)	762 (21.7%)
45–54	792 (10.6%)	296 (8.4%)
Place of residence		
Urban	1661 (22.2%)	1057 (30.1%)
Rural	5817 (77.8%)	2457 (69.9%)
Educational attainment		
No education	399 (5.3%)	279 (7.9%)
Primary	4252 (56.9%)	2175 (61.9%)
Secondary/higher	2827 (37.8%)	1060 (30.2%)
Wealth status		
Poor	2365 (31.6%)	1162 (33.1%)
Middle	1443 (19.3%)	688 (19.6%)
Rich	3670 (49.1%)	1664 (47.3%)
Religion		–
Christian	6560 (87.7%)	–
Muslim	726 (9.7%)	–
Traditional	192 (2.6%)	–
Marital status		
Never married	2932 (39.2%)	1580 (45.0%)
Married/cohabiting	4284 (57.3%)	1768 (50.3%)
Widowed	36 (0.5%)	17 (0.5%)
Divorced/Separated	226 (3.0%)	149 (4.2%)
Occupational status		
Not working	1089 (14.6%)	408 (11.6%)
Working	6389 (85.4%)	3106 (88.4%)
Access to media information		
No	1677 (22.4%)	440 (12.5%)
Yes	5801 (77.6%)	3074 (87.5%)
Contraceptive knowledge		
Poor	40 (0.5%)	76 (2.2%)
Good	7438 (99.5%)	3438 (97.8%)
Sex of household head		
Male head	6316 (84.5%)	3029 (86.2%)
Female head	1162 (15.5%)	485 (13.8%)

**Table 3 ijerph-20-05053-t003:** Bi-variate analysis of male involvement in FP decisions in Malawi and Tanzania.

Socio-Demographic Factors	Malawi		Tanzania	
	No	Yes	No	Yes
Age (in years)	*p* = 0.000; χ^2^ = 149.1174 *		*p* = 0.000; χ^2^ = 95.7591 *	
15–24	1769 (54.8)	1457 (45.2)	1265 (81.3)	291 (18.7)
25–34	785 (39.8)	1190 (60.3)	599 (66.6)	301 (33.4)
35–44	597 (40.2)	888 (59.8)	499 (65.5)	263 (34.5)
45–54	364 (46.0)	428 (54.0)	215 (72.6)	81 (27.4)
Place of residence	*p* = 0.065; χ^2^ = 4.4266		*p* = 0.000; χ^2^ = 26.1499 *	
Urban	743 (44.7)	918 (55.3)	714 (67.6)	343 (32.5)
Rural	2772 (47.7)	3045 (52.4)	1864 (75.9)	593 (24.1)
Educational attainment	*p* = 0.000; χ^2^ = 62.4618 *		*p* = 0.102; χ^2^ = 21.1030	
No education	212 (53.1)	187 (46.9)	235 (84.2)	44 (15.8)
Primary	2138 (50.3)	2114 (49.7)	1595 (73.3)	580 (26.7)
Secondary/higher	1165 (41.2)	1662 (58.8)	748 (70.6)	312 (29.4)
Wealth status	*p* = 0.404; χ^2^ = 11.2283		*p* = 0.005; χ^2^ = 34.0326 *	
Poor	1165 (49.3)	1200 (50.7)	922 (79.4)	240 (20.7)
Middle	696 (48.2)	747 (51.8)	499 (72.5)	189 (27.5)
Rich	1654 (45.1)	2016 (54.9)	1157 (69.53)	507 (30.5)
Religion	*p* = 0.002; χ^2^ = 12.1411 *		-	
Christian	3036 (46.3)	3524 (53.7)	-	-
Muslim	373 (51.4)	353 (48.6)	-	-
Traditional	106 (55.2)	86 (44.8)	-	-
Marital status	*p* = 0.001; χ^2^ = 111.6213 *		*p* = 0.005; χ^2^ = 55.7393 *	
Never married	1592 (54.3)	1340 (45.7)	1247 (78.9)	333 (21.1)
Married/cohabiting	1828 (42.7)	2456 (57.3)	1234 (69.8)	534 (30.2)
Widowed	19 (52.8)	17 (47.2)	9 (52.9)	8 (47.1)
Divorced/Separated	76 (33.6)	150 (66.4)	88 (59.1)	61 (40.9)
Occupational status	*p* = 0.080; χ^2^ = 178.0138		*p* = 0.005; χ^2^ = 48.8549 *	
Not working	715 (65.7)	374 (34.3)	358 (87.8)	50 (12.25)
Working	2800 (43.8)	3589 (56.2)	2220 (71.5)	886 (28.5)
Access to media information	*p* = 0.001; χ^2^ = 29.4754 *		*p* = 0.902; χ^2^ = 56.5174	
No	886 (52.8)	791 (47.2)	388 (88.2)	52 (11.8)
Yes	2629 (45.3)	3172 (54.7)	2190 (71.2)	884 (28.8)
Contraceptive knowledge	*p* = 0.000; χ^2^ = 45.3407 *		*p* = 0.006; χ^2^ = 28.2035	
Poor	40 (100.0)	00 (100.0)	76 (100.0)	00 (100.0)
Good	3475 (46.7)	3963 (53.3)	2502 (72.8)	936 (27.3)
Sex of household head	*p* = 0.004; χ^2^ = 13.1997 *		*p* = 0.057; χ^2^ = 3.6154	
Male	2912 (46.1)	3404 (53.9)	2205 (72.8)	824 (27.2)
Female	603 (51.9)	559 (48.1)	373 (76.9)	112 (23.1)

Ref., Reference category; * *p* < 0.05.

**Table 4 ijerph-20-05053-t004:** Multivariate analysis of male involvement in FP decisions in Tanzania and Malawi.

Sociodemographic Factors	Malawi		Tanzania	
Age (in years)	UOR	AOR	UOR	AOR
15–24	RC	RC	RC	RC
25–34	1.47 (1.26–1.72)	1.84 (1.64–2.06)	1.89 (1.47–2.44)	2.18 (1.81–2.64)
35–44	1.47 (1.24–1.75) *	1.81 (1.59–2.05) **	2.00 (1.50–2.67)	2.29 (1.88–2.79)
45–54	1.25 (1.03–1.53) *	1.43 (1.22–1.67) **	1.49 (1.04–2.13)	1.64 (1.23–2.18)
Place of residence				
Urban	RC	RC	RC	RC
Rural	0.98 (0.86–1.12)	0.89 (0.80–0.99)	0.79 (0.65–0.96) *	0.66 (0.57–0.78) *
Educational level				
No education	RC	RC	RC	RC
Primary	1.25 (1.01–1.55)	1.12 (0.91–1.38)	1.90 (1.34–2.68) ***	1.94 (1.39–2.72) **
Secondary/higher	1.72 (1.37–2.15) **	1.62 (1.31–1.99) ***	2.35 (1.61–3.42)	2.23 (1.57–3.16)
Wealth status				
Poor	RC	RC	RC	RC
Middle	1.01 (0.89–1.16)	1.04 (0.91–1.19)	1.26 (1.00–1.58) ***	1.46 (1.17–1.81) ***
Rich	1.06 (0.93–1.20) **	1.18 (1.07–1.31) **	1.17 (0.94–1.47)	1.68 (1.41–2.01)
Religion				
Christian	RC	RC	RC	RC
Muslim	0.86 (0.74–1.01) *	0.82 (0.70–0.95) *		
Traditional	0.74 (0.55–1.00) *	0.70 (0.52–0.93) *		
Marital status				
Never married	RC	RC	RC	RC
Married/cohabiting	1.04 (0.89–1.22)	1.60 (1.45–1.75)	0.94 (0.73–1.21) *	1.62 (1.38–1.90) *
Widowed	0.74 (0.38–1.47)	1.06 (0.55–2.05)	1.99 (0.71–5.62)	3.33 (1.27–8.70)
Divorced/Separated	1.56 (1.15–2.13)	2.34 (1.76–3.12)	1.60 (1.08–2.39)	2.60 (1.83–3.68)
Occupational status				
Not working	RC	RC	RC	RC
Working	2.07 (1.78–2.40)	2.45 (2.14–2.80)	2.33 (1.67–3.26) **	2.86 (2.10–3.88) **
Access to media information				
No	RC	RC	RC	RC
Yes	1.18 (1.06–1.33) *	1.35 (1.21–1.51) *	2.45 (1.79–3.33)	3.01 (2.23–4.06)
Sex of household head				
Male	RC	RC	RC	RC
Female	1.02 (0.89–1.17) **	1.79 (1.70–1.90) *	0.91 (0.71–1.15) **	0.80 (0.64–1.01) *

* *p* < 0.001; ** *p* < 0.01, *** *p* < 0.05; RC = Reference category.

## Data Availability

The dataset, which contains data from the Demographic and Health Surveys, is freely available to qualified researchers. A written request was made to the DHS MACRO in order to acquire the DHS Measure data, and authorization was given to utilize the data for this study. Please apply at https://dhsprogram.com/data/dataset_ad-min/loginmain.cfm (accessed on 14 June 2022) to seek access to the dataset.

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
