# Peer review of "Male Involvement in Family Planning Decisions in Malawi and Tanzania: What Are the Determinants?"

_ijerph, 2023, doi:10.3390/ijerph20065053_

Round 1
Reviewer 1 Report
Authors indicated in the Conclusions section "Therefore, government 480 and civil society organizations, FP programme planners, and different stakeholders 481 should ensure availability, accessibility, and sustained advocacy for male participation in 482 FP decision-making and the effective utilization of FP services, in addition to creating a 483 conducive environment for the dissemination of knowledge on the huge benefits of male 484 participation in FP decisions among couples, behavioural change, and open discussion on 485 the issue of FP with their spouses relating to reproductive health issues, including FP 486 methods and services. "
This a important and central theme resulting from the findings. However, the description above needs more details, or examples, or solutions/strategies for practitioners to elicit behavior change in male target audiences.
These inclusions could be added, maybe more appropriately, to the Discussion section.
The methodology is robust and substantive. A greater connection, with details, from research to practice or "how does the scholarship inform practice?" necessitates expansion with solutions for those administering the recommended programs. This article https://doi.org/10.3390/ijerph19084577 might serve as a guide for you in addressing my assessment of those last sections in the manuscript.
Overall, a well-done piece of scholarship that just needs more specificities in the Discussion and/or Conclusions.
Author Response
Authors’ Responses to Reviewer 1
Authors indicated in the Conclusions section "Therefore, government 480 and civil society organizations, FP programme planners, and different stakeholders 481 should ensure availability, accessibility, and sustained advocacy for male participation in 482 FP decision-making and the effective utilization of FP services, in addition to creating a 483 conducive environment for the dissemination of knowledge on the huge benefits of male 484 participation in FP decisions among couples, behavioural change, and open discussion on 485 the issue of FP with their spouses relating to reproductive health issues, including FP 486 methods and services. "
This an important and central theme resulting from the findings. However, the description above needs more details, or examples, or solutions/strategies for practitioners to elicit behavior change in male target audiences. [Appropriately addressed in Green ink colour in the main manuscript]
These inclusions could be added, maybe more appropriately, to the Discussion section. [The inclusion has been moved to the Discussion section and appropriately addressed it according to the reviewer suggestions in Green ink colour in the main manuscript]
The methodology is robust and substantive. A greater connection, with details, from research to practice or "how does the scholarship inform practice?" necessitates expansion with solutions for those administering the recommended programs. This article https://doi.org/10.3390/ ijerph19084577 might serve as a guide for you in addressing my assessment of those last sections in the manuscript. [This has been adopted and reference cited also in Green ink colour in the main manuscript]
Overall, a well-done piece of scholarship that just needs more specificities in the Discussion and/or Conclusions. [All the specificities in the Discussion and/or Conclusions has been appropriately and adequately addressed in Green ink colour in the main manuscript]

Reviewer 2 Report
Male involvement in family planning is an important issue. This paper applies DHS data from Malawi and Tanzania and searches for determinants of male involvement in family planning decision and contraceptive use. Although the findings are interesting and knowledgeable, some issues should be addressed or explained:
(1) What is the main reason to compare the two countries? It seems that the prevalence of male involvement and social-economic determinants differ between the two countries.
(2) A clear framework for multivariate analysis is missing, and the selection of independent variables should be better explained.
(3) Male involvement related to discussion and negotiation of family planning issues between the partners, therefore social-economic status of women partner is equally important. This point is missing in the study.
(4) The structure of the paper is sometimes confused, as there is conclusion in the “discussion” section, and there are discussions in “conclusion” section.
Author Response
Authors’ Responses to Reviewer 2
Male involvement in family planning is an important issue. This paper applies DHS data from Malawi and Tanzania and searches for determinants of male involvement in family planning decision and contraceptive use. Although the findings are interesting and knowledgeable, some issues should be addressed or explained:
- What is the main reason to compare the two countries? It seems that the prevalence of male involvement and social-economic determinants differ between the two countries. [The reasons for the two countries comparison is inserted in page 2 and 3 in blue ink colour]
- A clear framework for multivariate analysis is missing, and the selection of independent variables should be better explained. [This has been inserted in Page 5 in blue ink colour]
- Male involvement related to discussion and negotiation of family planning issues between the partners, therefore social-economic status of women partner is equally important. This point is missing in the study. [This has been inserted in Page 12 in blue ink colour]
- The structure of the paper is sometimes confused, as there is conclusion in the “discussion” section, and there are discussions in “conclusion” section. [This has been addressed, as some part of the discussion in the conclusion has been moved to the discussion section. See the insertions in green ink colour]

Round 2
Reviewer 1 Report
The authors greatly improved the manuscript and addressed each of my edits and recommendations.